# Effects of Gluten on Gut Microbiota in Patients with Gastrointestinal Disorders, Migraine, and Dermatitis

**DOI:** 10.3390/nu16081228

**Published:** 2024-04-20

**Authors:** Ismael San Mauro Martín, Sara López Oliva, Elena Garicano Vilar, Guerthy Melissa Sánchez Niño, Bruno F. Penadés, Ana Terrén Lora, Sara Sanz Rojo, Luis Collado Yurrita

**Affiliations:** 1Research Centers in Nutrition and Health (CINUSA Group), Paseo de la Habana 43, 28036 Madrid, Spain; info@grupocinusa.es (I.S.M.M.); sara@grupocinusa.es (S.L.O.); elena@grupocinusa.es (E.G.V.); melissa@grupocinusa.es (G.M.S.N.); estadistica@grupocinusa.es (B.F.P.); ana.terren@grupocinusa.es (A.T.L.); sara.sanz@grupocinusa.es (S.S.R.); 2Department of Medicine, Universidad Complutense de Madrid, 28040 Madrid, Spain

**Keywords:** gluten, microbiota, fungi, bacteria, zonulin, gastrointestinal disorders, MIDAS

## Abstract

As gluten may trigger gastrointestinal disorders (GIDs), its presence or absence in the diet can change the diversity and proportion of gut microbiota. The effects of gluten after six weeks of a double-blind, placebo-controlled intervention with a gluten-free diet (GFD) were studied in participants with GIDs suffering from migraines and atopic dermatitis (n = 46). Clinical biomarkers, digestive symptoms, stool, the Migraine Disability Assessment questionnaire, and zonulin levels were analyzed. Next-generation sequencing was used to amplify the 16S rRNA gene of bacteria and the internal transcribed spacer (ITS) regions of fungi. The GFD increased Chao1 fungal diversity after the intervention, while the fungal composition showed no changes. Bacterial diversity and composition remained stable, but a positive association between bacterial and fungal Chao1 diversity and a negative association between Dothideomycetes and *Akkermansia* were observed. GIDs decreased in both groups and migraines improved in the placebo group. Our findings may aid the development of GID treatment strategies.

## 1. Introduction

Gastrointestinal (GI) disorders are a common phenomenon, and individuals often report that the onset or worsening of symptoms occurs after consuming a specific food that causes discomfort and seek dietary advice from their physician. In this context, irritable bowel syndrome, one of the most common gastrointestinal disorders (GIDs) linked to disorders of the gut–brain axis, occurs in up to 84% of patients who report GI distress and may be caused by consumption of at least one trigger food [1].

The role of diet in GIDs has recently gained attention, and new trends include the low-carbohydrate, low-fermentable oligosaccharide, disaccharide, monosaccharide, and polyol (FODMAP) diet, hypoallergenic diets, and the gluten-free diet (GFD) [2]. The GFD has become the cornerstone treatment for celiac disease (CD) and its use has spread to many other gastrointestinal and extraintestinal conditions, with a cumulative prevalence of approximately 10% [3]. Symptoms of gluten intolerance include abdominal pain, irregular bowel movements, muscle aches, headaches, and fatigue. These symptoms improve with GFD but reappear rapidly after consuming gluten-containing grains. However, the causal role of gluten remains unclear [4], even after double-blind, placebo-controlled studies [5,6].

Other wheat components, such as carbohydrates or amylase–trypsin inhibitors (ATIs), may be partly responsible for symptoms of gluten sensitivity, since gluten-containing grains have been shown to contain large quantities of ATIs and are very resistant to food processing and digestion [7] in addition to their ability to stimulate the immune system by activating myeloid cells in the intestine that can cause or maintain intestinal inflammation [8].

Based on previous studies, gluten could regulate molecules that participate in intestinal permeability, such as zonulin. In addition, if gluten is not correctly hydrolyzed, its peptides can be absorbed and provide an immune-inflammatory reaction [9], increasing levels of Toll-like receptor 2 (TLR2), IFN-γ, and intraepithelial CD3+ T-cell lymphocytes [10].

Some extraintestinal symptoms related to the inclusion of gluten in the diet are migraine and atopic dermatitis. Migraine is a debilitating headache that affects 20% of the world population. It is defined as a recurrent headache of medium to severe intensity lasting 4–72 h, accompanied by nausea and vomiting as GI symptoms and is presumably influenced by gut microbiota due to the microbiota–gut–brain axis [11]. A dysfunction in this gut–brain axis can aggravate the effect of gluten as an inflammatory modulator [12,13] because a dysbiosis in gut microbiota can cause systemic inflammation facilitating extraintestinal symptoms [10]. In addition, dermatitis is another extraintestinal manifestation related to the consumption of gluten or allergenic foods and presents as a rash with itching and blisters [14]. Patients with these symptoms can often make dietary changes, and the elimination of gluten and wheat shows more than 50% improvement in their symptoms [15].

In addition to these symptoms, gut microbiota have gained increasing interest because they are considered the basis of a healthy gut, and dysbiosis is associated with several metabolic and inflammatory diseases [16,17]. Therefore, gut microbiota play a crucial role in the pathophysiology of gluten-related disorders [4]. In fact, some studies show that bacteria related to protective effects decreased in patients with CD (e.g., *Bifidobacteria*, *Firmicutes*, *Lactobacilli*, *Streptococceae*), whereas harmful Gram-negative bacteria increased (e.g., *Bacteroides*, *Bacterioidetes*, *Bacteroides fragilis*, *Prevotella*, *E. Coli*, *Proteobacteria*, *Haemophilus*, *Serratia*, *Klebsiella*) [18].

However, the effect of gluten on inflammatory response or microbiota composition is not clear yet. Therefore, the objective of this study was to investigate the effect of gluten on gut microbiota composition and diversity, symptoms, intestinal permeability, and biomarkers after six weeks of a double-blind, placebo-controlled intervention with a GFD in individuals with GIDs suffering from migraine and atopic dermatitis.

## 2. Materials and Methods

### 2.1. Participants

The study employed 46 participants with GIDs and assigned them to two study groups. The gluten group was composed of 17 participants who consumed 8 g of gluten in sachets daily, and the placebo group consisted of 29 participants who consumed placebo in sachets daily. The characteristics of both groups were homogeneous except for the occurrence of nausea (Table 1). In addition, a proportion of participants with migraine (n = 26, 56.5%) and atopic dermatitis (n = 20, 43.5%) were identified and assessed.

### 2.2. Study Design

A randomized, double-blind, placebo-controlled clinical trial was designed to investigate the effect of gluten on gut microbiota composition and diversity, symptoms, intestinal permeability, and biomarkers in 46 participants with GIDs suffering from migraine (n = 26) and dermatitis (n = 20) who followed a gluten free diet, consuming 8 g of gluten or placebo in sachets daily for 6 weeks (Figure 1). The diet was administered by expert dietitians with semi-personalized and standardized menus and was normocaloric and based on a Mediterranean diet.

For the development of the placebo or gluten sachets, they were manufactured taking into account the binding characteristics of gluten and had a pleasant flavor to ingest dissolved in water. Both had similar appearance, taste, color, smell, and binding properties. For this, the gluten challenge kit (Quaonlab, Spain) was used, which is based on the Salerno experts’ criteria [19] of diagnosis for non-celiac gluten sensitivity (NCGS) with the intake of 8 g/day of gluten.

Data on clinical biomarkers, GI symptoms, and stool were collected, and the MIDAS questionnaire was applied to quantify migraine-related disability. An intestinal permeability test was performed to measure zonulin levels. The study was carried out by the Research Centers in Nutrition and Health (CINUSA) Group in Madrid, Spain, at the CINUSA Clinic and the Ruben International Hospital (Paseo de la Habana, Madrid, Spain).

All patients included in the study satisfied the inclusion and exclusion criteria. All participants signed the informed consent before enrollment, and their privacy was protected. The inclusion criteria included (1) patients aged 18 to 60 years old; (2) patients with gastrointestinal disorders and migraine or atopic dermatitis; (3) patients with GIDs, understood as the presence of at least 2 of the following symptoms six months earlier: bloating, abdominal pain, flatulence, diarrhea, borborygmi, constipation, defecatory urgency, incomplete evacuation, nausea, burning, burps, regurgitation, or epigastric pain. The exclusion criteria were as follows: (1) patients with chronic obstructive pulmonary disease, cancer, lupus, AIDS, stroke, hepatitis, tuberculosis, sclerosis, colectomies, enterectomies, ulcerative colitis, Crohn’s disease, celiac disease, and wheat allergy; (2) patients who follow a gluten-free diet.

### 2.3. Biomarker Analysis

Blood and stool samples were collected at the beginning (before intervention) and at the end of the study (when treatment was finalized). Blood samples were collected by nursing personnel according to the standard venipuncture procedure.

Blood samples were collected from study participants to analyze biomarkers such as iron, ferritin, and transferrin using PCR. In addition, fecal zonulin levels were quantified as a biomarker of intestinal permeability using human zonulin ELISA kits at the start and end of the intervention by orally administering non-metabolizable substances [20,21].

### 2.4. MIDAS Questionnaire

The MIDAS questionnaire was employed, consisting of seven questions in total, including three questions assessing the number of days lost due to migraine at school/work, household chores, and family/leisure activity domains (MIDAS items 1, 3, and 5) and two questions assessing the number of additional days with limited productivity due to migraine at school/work and household chore domains (MIDAS items 2 and 4). The total MIDAS score is the sum of the days given in response to these five questions (MIDAS 1 to MIDAS 5) and ranges from 0 to 90. This questionnaire is used to categorize patients into disability grades I to IV. A higher score indicates a more severe disability [22].

### 2.5. Molecular Analysis

#### 2.5.1. DNA Extraction

DNA was extracted from fecal samples using the PSP Spin Stool DNA Plus Kit (Invitek Molecular GmbH, Berlin, Germany), following the manufacturer’s instructions. Protocol 3 was followed to isolate total DNA of difficult-to-lyse bacteria from 1.4 mL of fecal homogenate. After collection in a 2.0 mL Safe-Lock tube, the homogenized sample was incubated for 10 min at 95 °C in a thermomixer under continuous shaking at 900 rpm. The samples were then incubated on ice for 3 min and placed in a thermoblock at 95 °C for 3 min. Next, five Zirconia Beads II (Invitek Molecular GmbH, Berlin, Germany) were added to the homogenate and shaken for 2 min at 20 °C. The samples were then centrifuged at 12,000 rpm for 1 min. To remove PCR inhibitors, the supernatant was transferred to an InviAdsorb tube (Invitek Molecular GmbH, Berlin, Germany), and the suspensions were incubated for 1 min at RT, followed by a second sample clean-up and digestion by transferring 25 µL of proteinase K and 800 µL of the supernatant to a new 1.5 mL tube, vortexing, and incubating at 70 °C for 10 min with continuous shaking at 900 rpm. DNA binding was performed by adding 400 µL of P-binding buffer to the lysate, briefly vortexing to transfer the samples to the RTA Spin Filter membrane (Invitek Molecular GmbH, Berlin, Germany) and incubating at RT for 1 min. The samples were then centrifuged at 10,000 rpm for 2 min, followed by washing and ethanol removal. Finally, DNA was eluted by placing the RTA Spin Filter in a new 1.5 mL tube and adding 100–200 µL of preheated elution buffer D, incubating at 70 °C for 3 min, and centrifuging at 10,000 rpm for 1 min.

#### 2.5.2. DNA Quality

DNA concentration was measured with the Quant-iT PicoGreen dsDNA Assay Kit (Thermo Fisher Scientific, Waltham, MA, USA) using the VICTOR^3^ Multilabel Plate Reader (PerkinElmer, Waltham, MA, USA) at 260 nm. Fluorescence-based quantification was used to quantify the DNA using a specific dye for double-stranded DNA, which can be specifically and accurately quantified even in the presence of many common contaminants.

#### 2.5.3. Library Construction

The sequencing library was prepared by random fragmentation of the DNA or complementary DNA sample, followed by 5′ and 3′ primer ligation. Alternatively, fragmentation and ligation were combined in a single step since this greatly increases the efficiency of the library construction process. The adapter-ligated fragments were amplified by PCR and purified using agarose gel electrophoresis.

#### 2.5.4. Next-Generation Sequencing

For cluster generation, the library was loaded into a flow cell where the fragments were captured using surface-bound oligos complementary to the library adapters. Each fragment was then amplified into distinct clonal clusters by bridge amplification.

When cluster generation was complete, the templates were ready for sequencing, which was performed using MiSeq high-throughput sequencing (Illumina, San Diego, CA, USA) to examine gut bacterial and fungal communities by PCR amplification of the V4 region of the 16S rRNA and ITS1 and ITS2 of the internal transcribed spacer (ITS) regions.

### 2.6. Bioinformatics Analysis

The USEARCH v11.1 software [23] was used for bioinformatics processing of the sequenced samples. This software analyzes the 16S (bacteria) and ITS (fungi) rRNA gene amplicon sequences using various heuristic computational algorithms to determine the zero-radius operational taxonomic units (ZOTUs) in the samples and estimate their abundance in each sample and their corresponding taxonomic annotation. The quality of the sequences was assessed before starting the bioinformatics analysis.

The initial number of ZOTUs was 1671 (3,253,958 reads) in the 16S rRNA gene amplicon. After excluding non-prokaryote-associated sequences (1 ZOTU) and filtering ZOTUs of very low abundance (n ≤ 10; 105 ZOTUs), a total of 1565 ZOTUs and 3,251,687 effective reads were obtained. Regarding the ITS amplicon, 1615 ZOTUs (4,979,818 reads) were found, and 4,979,435 effective reads were obtained after excluding ZOTUs with very low abundance (n ≤ 10; 561 ZOTUs). Metagenomic data were never rarefied since sequencing, clustering, and taxonomic assignment were correct.

Taxonomic assignment was very poor downstream of the class level in the ITS amplicon. While phylum and class levels were around 55% on average, the rest hardly reached 20% and the genus level was below 10%. It was therefore decided that fungal classes will be used as the highest level since the number of readings was insufficient and the representation was evident in the other levels. In contrast, data loss was low for the 16S rRNA gene amplicon, with taxonomic assignment to genus reaching 80%.

### 2.7. Statistical Analysis

The normality of the distribution of the independent samples and differences for paired samples were examined using the Shapiro–Wilk test. Symmetry was examined in the last case with the standardized skewness. Paired samples were compared using the paired Student’s *t*-test when the distribution of the differences was normal or the Wilcoxon signed-rank test when the distribution was not normal or the sign test in the absence of symmetry. The Student’s *t*-test was used for unpaired samples and the MW test was used for biased distributions. The proportions of the initial sample data with categorical variables were compared using the Pearson’s exact chi-squared test.

Alpha diversity was analyzed using the Chao1 and Shannon indices, with comparison of the study groups using time differences (∆ = Final − Basal), and beta diversity was analyzed using PERMANOVA, with results visualized with PCoA. Linear regressions of Chao1 diversity between fungi and bacteria were also reported and heatmaps were created by adding Spearman rank-order correlations to search for associations between biomarkers and the most abundant bacterial genera as well as between fungi and bacteria. Most analyses were performed using the microbial community analysis package microeco (v0.12.0) [24].

Fungal and bacterial composition were assessed by proposing a mixed-effects model, with subjects included as random effects and time, group, sex, age, and BMI included as fixed effects. Relative abundance was added to the model, and values with a minimum prevalence of 10% were normalized with total-sum scaling and transformed with arcsine square-root using the package MaAsLin2 (v1.8.0) to search for multivariate associations with linear models [25].

Taxa without taxonomic assignment were excluded from the analysis in all tests except the alpha and beta diversity analyses. All contrasts in the study were bilateral and the level of significance was corrected for multiple contrasts (FDR) when appropriate with a 95% confidence interval. The tests and visualizations of the present study were performed using the statistical software R (v4.1.2).

### 2.8. Ethical Considerations

The study was approved by the Ethics Committee for Research with Medicines (Comité Ético de Investigación con Medicamentos) of the Madrid Ruber International Hospital (QuirónSalud Group, Madrid, Spain) (Sa-16151/19—EC: 393; CIN-GLU-01-19).

The study was conducted in compliance with the latest version of the Declaration of Helsinki (World Medical Association, 2013), good clinical practice standards (ICH 2016 R2), and legal standards and regulations for biomedical research in humans (Law 14/2007 and royal decree 1090/2015).

In addition, the study was conducted under strict compliance with the European General Data Protection Regulation 2016/679 of the European Parliament and EU Regulation 2016/679 of 27 April 2016, on data protection of natural persons (RGPD) and Organic Law 3/2018 (LOPD).

All participants took part in the study voluntarily. Only those who met the inclusion criteria and signed the informed consent form were part of the study.

## 3. Results

### 3.1. Diversity and Fungal Composition

Alpha diversity indices revealed that the gluten group shows an important, although not significant, reduction in Chao1 diversity (n = 15, paired *t*-test, *p* = 0.058; Figure 2A), although the Shannon index remained unchanged (n = 15, paired *t*-test, *p* = 0.812). The Chao1 indices (n = 24, paired *t*-test, *p* = 0.011) increased in the placebo group and decreased in the gluten group. Time differences (∆ = Final − Basal) were significantly higher when comparing the two study groups (*t*-test, *p* = 0.002). Moreover, the Shannon index was significantly increased in the placebo group (n = 24, paired *t*-test, *p* = 0.021), but there were no differences between groups (*t*-test, *p* = 0.191). Permutational multivariate analysis of variance (PERMANOVA) could not detect differences in fungal community distribution between groups (*p* > 0.700; Figure 2B).

Analysis of the relative abundance of fungal phyla of all participants with sequencing data showed that gut microbiota were almost entirely dominated by Ascomycota before intervention in both study groups (>95%, Figure 2C). The phylum Basidiomycota was negligible initially but increased slightly in both groups, whereas the phylum Mucoromycota was negligible in both groups. Therefore, no changes were found in the Basidiomycota/Ascomycota ratio after intervention in any group (Appendix A). Saccharomycetes was the predominant class followed by Eurotiomycetes in both groups before the intervention. Gut microbiota continued to be dominated by Saccharomycetes after the intervention, but Eurotiomycetes declined in both groups. Interestingly, Tremellomycetes decreased in the gluten group and Dothideomycetes decreased in the placebo group.

These changes were tested with the MaAsLin2 package to determine whether a significant differential abundance between study times and groups was indeed present using a mixed-effects model with time, group, sex, age, and body mass index (BMI) as fixed effects and the participants as random effects. The results did not confirm any differences in fungal class abundance after the intervention and between groups (false discovery rate [FDR] > 0.05). Furthermore, none of the covariates (sex, age, and BMI) were significant individually.

### 3.2. Bacterial Diversity and Composition

No changes in alpha diversity were observed in any case when comparing study times and groups. Both the Chao1 (n = 16, paired *t*-test, *p* = 0.148, Figure 3A) and Shannon (n = 16, paired *t*-test, *p* = 0.568) indices revealed no significant results in the gluten and placebo groups after the intervention (Chao1: n = 27, paired *t*-test, *p* = 0.742; Shannon: n = 27, sign test, *p* = 0.424). PERMANOVA also could not detect differential spatial distributions of fungal communities between groups (*p* > 0.700; Figure 3B).

The bacterial composition of gut microbiota was dominated and segregated by the phyla Bacteroidetes and Firmicutes in both study groups with very subtle changes after the intervention (Figure 3C). In fact, no changes were found in the Firmicutes/Bacteroidetes ratio in either of the groups (Appendix A). The most abundant genus was *Bacteroides*, followed by *Faecalibacterium*, *Prevotella*, *Eubacterium*, and *Alistipes*, among others that were below 5%. These values were very similar between the study groups and remained after the intervention. This was confirmed with the MaAsLin2 package by following the same guidelines as for fungi and no significant results were found.

### 3.3. Relationship between ITS and 16S rRNA Gene Amplicon Sequencing Data

Sequencing data for fungi and bacteria were used to investigate the relationship between the ITS and 16S rRNA gene amplicons. Chao1 diversity in the ITS/16S ratio was higher in the placebo group (Wilcoxon, *p* < 0.05, Figure 4A) when comparing the differences between the groups (Mann–Whitney [MW] test, *p* < 0.05). This implies that the diversity of fungal species increased in relation to the decrease in bacterial species in the placebo group.

In addition, the Chao1 index steadily increased at the end of the intervention when comparing fungal and bacterial communities in the placebo group (n = 24, cor. = 0.496, R^2^ = 0.246, *p* = 0.014; Figure 4B); the gluten group showed the same trend but without statistical significance (n = 15, cor. = 0.283, R^2^ = 0.08, *p* = 0.307).

Furthermore, the final data on fungal classes and bacterial genera correlated with each other. A negative association was observed between the class Dothideomycetes and the genus *Akkermansia* in the microbiota of the gluten group after adjusting for significance (Spearman, FDR < 0.05, Figure 4C), but no such association was found in the placebo group (Spearman, FDR > 0.05, Appendix A).

### 3.4. GI Disorders, Migraine, and Atopic Dermatitis

The next step was to understand the effect of gluten in the severity of GIDs, migraine, and atopic dermatitis. First, it was observed that the placebo group reported more positive changes in sleep quality after the intervention. However, GI symptoms, such as abdominal pain, bloating, flatulence, diarrhea, rumbling sounds, reduced stool consistency, and Bristol stool scale, decreased in both study groups. For urgency to defecate, nausea, and belching, improvement was reported only in the gluten group. Conversely, the placebo group showed improvement in incomplete evacuation and epigastric pain (Table 2).

The Migraine Disability Assessment (MIDAS) questionnaire scores and Three-Item Severity (TIS) score variables were compared to investigate whether migraine or atopic dermatitis diagnosed by a professional in 24 participants (the difference in the number of participants was due to missing data) and 20 participants, respectively, was associated with gluten consumption and thus changes in symptoms after the intervention. Migraine improved only in the placebo group (n = 12, paired *t*-test, *p* < 0.05; Figure 5A), although this change was not significant when the groups were compared based on differences (MW test, *p* = 0.178). Other variables related to migraine, such as pain intensity, were approximately equal in both groups after the intervention, but the frequency of migraines in the previous three months improved in the placebo group (n = 12, paired *t*-test, *p* < 0.037, Appendix A). Although not significant, the placebo group with atopic dermatitis showed a substantial reduction in their symptoms (n = 14, paired *t*-test, *p* = 0.051, Figure 5B), which was not observed in the gluten group (Appendix A). The same was found for other symptoms related to itchy skin and scalp and dry skin, including in the neck, chest, and the rest of the whole body.

The possible association between participant characteristics, the MIDAS, the TIS score, the Bristol scale, and biomarkers of the most abundant bacterial genera in the final phase was analyzed. Spearman’s correlations did not provide significant results after adjusting for significance in any of the study groups (FDR > 0.05, Appendix A).

### 3.5. Biomarkers

Finally, iron metabolism, proteins, intestinal permeability, and blood counts were analyzed in all participants. Only an increase in transferrin levels was observed in the placebo group, but this was not sufficient to differentiate it from the gluten group (Appendix A).

## 4. Discussion

Diet affects the composition and function of the gut microbiota and the health of the host, especially in patients suffering from food-related diseases. It has been shown that the consumption of gluten leads to the formation of various gluten peptides in the small intestine as it is a substrate for different microorganisms located in the duodenum, thus contributing to dysbiosis in patients with diseases associated with food allergies [26]. Therefore, it is important to investigate the mechanisms used by the body to metabolize nutrients and their association with the symptoms of various recurrent diseases. In this context, this study investigated the effect of gluten on gut microbiota composition and diversity, symptoms, intestinal permeability, and biomarkers after six weeks of a double-blind, placebo-controlled intervention with a GFD in individuals with GIDs suffering from migraines and atopic dermatitis.

Our results suggest that the GFD group changed the gut fungal composition and diversity (Figure 2), whereas the bacterial community was unchanged with no significant differences before and after the intervention in both the gluten and placebo groups. The bacterial composition was dominated by the phyla Bacteroidetes and Firmicutes in both study groups, with very subtle changes after the intervention (Figure 3). This is consistent with a study that investigated the effect of gluten sensitivity on the microbiota, with the principal coordinate analysis (PCoA) based on gut microbiota composition showing a prevalence of the phylum Bacteroides [27]. Other bacteria belonging to the phylum Firmicutes have also been implicated in gluten metabolism, mostly of the genus *Lactobacillus*, followed by *Streptococcus*, *Staphylococcus*, and *Clostridium* [28]. Another study showed similar results with no significant differences found before and after a GFD, although *Bifidobacterium*, *Lactobacillus*, and *Bifidobacterium longum* counts decreased while *Enterobacteriaceae* and *Escherichia coli* counts increased in the gut microbiota of healthy patients after a GFD [29]. Though several studies present conflicting findings, there are significant data supporting the occurrence of gut dysbiosis after a GFD [30,31]. In fact, according to other studies, following a GFD for at least four weeks is sufficient to observe changes in symptoms and gut microbiota [18], although for a complete restoration of gut microbiota function, at least one year of a GFD could be necessary [32]. Regarding fungal diversity, the relative abundance of fungal phyla in all participants showed that the gut microbiota were dominated almost entirely by Ascomycota before the intervention in both study groups, which is often the most abundant fungus in the GI tract [33]. The phylum Basidiomycota increased in both gluten and placebo groups, whereas the increase in the phylum Mucoromycota was negligible. Therefore, no changes were observed in the Basidiomycota/Ascomycota ratio after the intervention in any group, except for Saccharomycetes, which was the predominant fungal class before and after the intervention. This is consistent with the findings of the Human Microbiome Project, which sequenced the ITS2 region to characterize fungal communities, showing a high prevalence of *Saccharomyces*, *Malassezia*, and *Candida* in the human gut [33].

The gut microbiota has been characterized in various diseases and GIDs [34,35,36], but few studies have investigated its composition in the context of a GFD or gluten sensitivity. Sequencing data for fungi and bacteria were used to investigate the relationship between the two amplicons, showing that Chao1 diversity in the ITS/16S ratio was higher in the placebo group, even when comparing the differences between groups. This implies that the diversity of fungal species increased in the gluten-free group. This can be due to the GFD and the low-FODMAP diet, which is also low in carbohydrates and fermented foods [27]. A linear increase was found at the end of the intervention when using Chao1 to compare the fungal and bacterial communities of the placebo group. Although the gluten group followed the same trend, it was not statistically significant. Fungal classes correlated with bacterial genera in the final data and a negative association was observed among the class Dothideomycetes, for a wide diversity of lifestyles, and for the genus *Akkermansia* in the gut microbiota of the gluten group, with no association found in the placebo group, which could be due to the action of lactic acid bacteria as biocontrol agents against fungi [37]. The class Dothideomycetes is one of the largest, most diverse, and most common fungal classes that infect crops such as rice, wheat, corn, or bananas [38], whereas the genus *Akkermansia* is a gut symbiont that colonizes the intestinal mucosa and is a promising candidate as a probiotic because it improves metabolic functions and host immune responses due to its lactic acid production [39].

Gluten causing GI and extra-GI symptoms in patients who do not have CD and are not allergic to wheat is a recurrent clinical problem [40]. Consequently, our study sought to understand the effect of gluten on the severity of GIDs, migraine, and atopic dermatitis. Firstly, participants assigned to the placebo group reported more positive changes in sleep quality after the intervention. GI symptoms, such as abdominal pain, bloating, flatulence, diarrhea, rumbling sounds, reduced stool consistency, and Bristol stool scale, decreased in both study groups. This differs from a study conducted with 34 participants, in which 13 reported that GI symptoms were not adequately controlled compared with 6 of 15 participants in the placebo group who reported an improvement in GI symptoms. However, GI symptoms in the study group worsened substantially one week after the gluten intervention in terms of general symptoms (bloating, stool consistency, and fatigue) [41]. This was also observed in another study with 35 participants following a GFD in a double-blind challenge study, in whom GI symptoms were significantly worse after the gluten challenge compared to baseline (pain, reflux, indigestion, diarrhea, constipation) [5].

It should be noted that improvement in symptoms such as urgency to defecate, nausea, and belching was reported only in the gluten group. Conversely, incomplete evacuation and epigastric pain improved in the placebo group (Table 2).

The MIDAS scores and TIS variables were compared to investigate whether migraine or atopic dermatitis were associated with gluten consumption and to analyze changes in symptoms after the intervention in participants diagnosed with those conditions. Migraine only improved in the participants in the placebo group, although the association between headache, specifically migraine, and gluten has been better documented in patients with CD. Migraine is twice as common in patients with CD and gluten sensitivity compared with controls [42]. However, this improvement in migraine was not statistically significant when the groups were compared based on differences. Other migraine-related variables, such as pain intensity, were equally intense in both groups after the intervention, but the frequency in the previous 3 months improved in the placebo group. Moreover, although not significant, participants in the placebo group with atopic dermatitis had their symptoms substantially reduced, which was not observed in the gluten group. A study investigating dermatitis herpetiformis (DH) and its association with gluten consumption showed that a GFD does not affect DH [43]. However, some studies show that up to 18% of patients with DH acquire tolerance to gluten during a GFD and do not relapse after the reintroduction of gluten [44,45,46,47]. Finally, iron metabolism, proteins, intestinal permeability, and blood counts were analyzed in all participants, and an increase in transferrin levels was observed in the placebo group, although this change was not sufficient to distinguish it from the gluten group. In general, individuals reported a greater deterioration of their well-being with gluten consumption compared with the placebo group.

## 5. Conclusions

Our data revealed a decrease in the overall well-being and quality of life of individuals with functional GI symptoms while unknowingly consuming gluten, thereby confirming that GI and extra-GI symptoms were triggered by gluten and that a GFD can have a beneficial effect on health. Moreover, individuals who respond to a GFD showed a symptomatic relapse during a blind placebo-controlled gluten challenge, thereby confirming that gluten consumption can trigger GID symptoms and impair quality of life.

## Figures and Tables

**Figure 1 nutrients-16-01228-f001:**
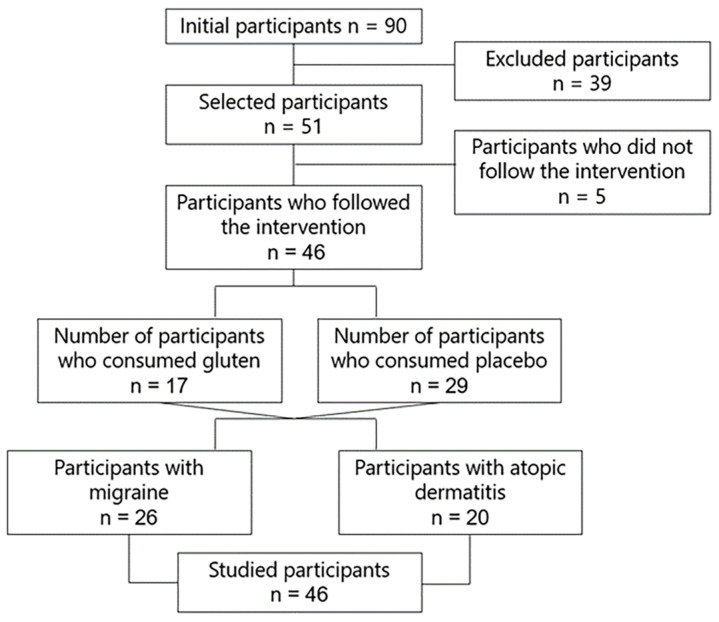
Study flowchart.

**Figure 2 nutrients-16-01228-f002:**
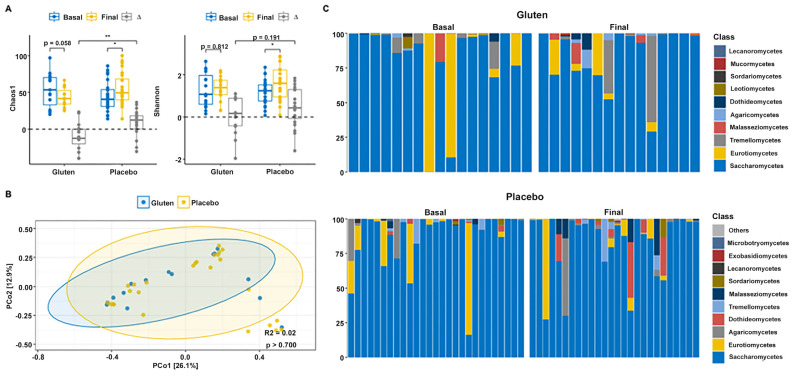
Gut fungal diversity and composition of participants according to study group. (**A**) Chao1 (left) and Shannon (right) alpha diversity indices in the gluten (n = 15) and placebo (n = 24) groups. (**B**) Principal coordinate analysis (PCoA) based on the Bray–Curtis distance. (**C**) Relative abundance of the 10 dominant fungal classes in the gluten (top) and placebo (bottom) groups according to basal and final time. ∆ = Final − Basal. * *p* < 0.05; ** *p* < 0.01.

**Figure 3 nutrients-16-01228-f003:**
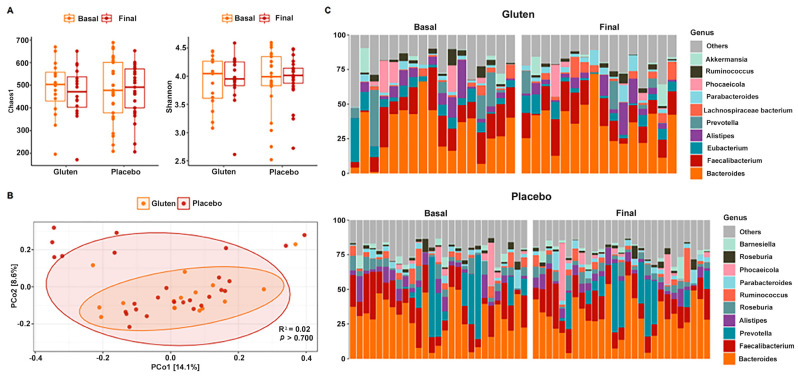
Gut bacterial diversity and composition of participants according to study group. (**A**) Chao1 (left) and Shannon (right) alpha diversity indices in the gluten (n = 16) and placebo (n = 27) groups. (**B**) PCoA based on the Bray–Curtis distance. (**C**) Relative abundance of 10 dominant bacterial genera in the gluten (top) and placebo (bottom) groups according to basal and final time.

**Figure 4 nutrients-16-01228-f004:**
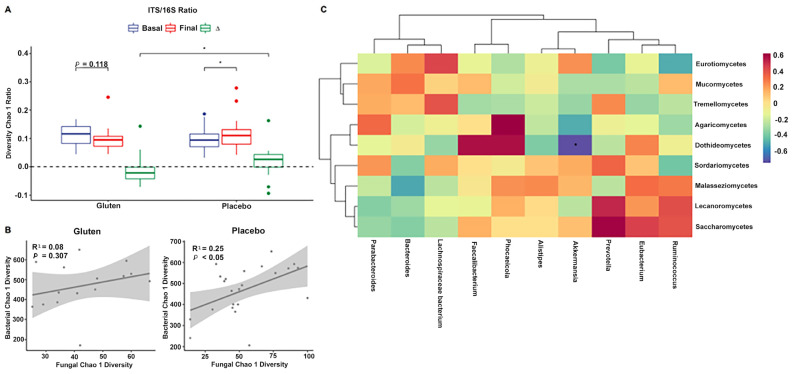
Relationship between fungi and bacteria after the intervention. (**A**) Internal transcribed spacer (ITS)/16S ratio of Chao1 richness index. (**B**) Linear regression of fungal and bacterial Chao1 diversity indices in the gluten (left, n = 15) and placebo (right, n = 24) groups. (**C**) Hierarchical clustering of correlations using the Spearman’s test between bacterial genera (bottom) and fungal classes (right side) in the gluten group. ∆ = Final − Basal. * *p* < 0.05 (ITS/16S), * false discovery rate < 0.05 (heatmap).

**Figure 5 nutrients-16-01228-f005:**
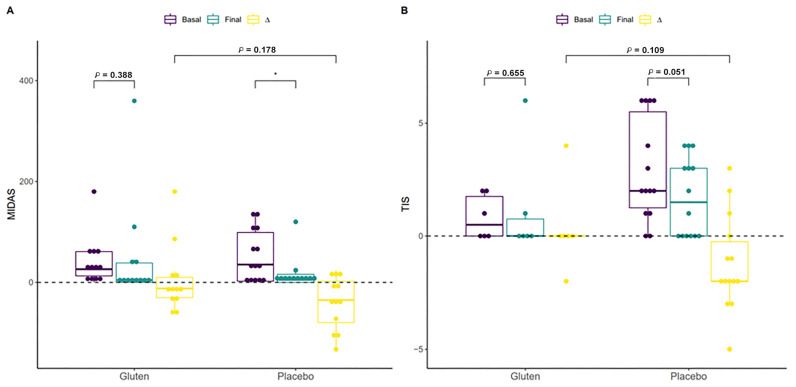
Effect of the intervention on participants with migraine or atopic dermatitis. (**A**) Effect of gluten (n = 12) and placebo (n = 12) on Migraine Disability Assessment (MIDAS) among participants with migraine. (**B**) Effect of gluten (n = 6) and placebo (n = 14) on the Three-Item Severity (TIS) score among participants with atopic dermatitis. ∆ = Final − Basal. * *p* < 0.05.

**Table 1 nutrients-16-01228-t001:** Initial characteristics of participants according to study group.

	Gluten Group (n = 17)	Placebo Group (n = 29)	*p*
**Age** (mean ± standard deviation)	41.59 ± 9.24	41.28 ± 8.93	0.910
**Sex, n (%)**			
Women	11 (64.7)	25 (86.2)	0.139
Men	6 (35.3)	4 (13.8)	
**Body mass index,** (median (interquartile range))	23.8 (21.4–27.8)	24.6 (22.95–27.05)	0.482
**Gastrointestinal (GI) symptoms** **, n (%)**			
Abdominal pain	15 (88.2)	22 (75.9)	0.450
Bloating	12 (70.6)	26 (89.7)	0.125
Flatulence	15 (88.2)	27 (93.1)	0.619
Diarrhea	9 (52.9)	13 (44.8)	0.761
Stomach sounds	11 (64.7)	24 (82.8)	0.282
Constipation	6 (35.3)	11 (37.9)	1
Urge to defecate	9 (52.9)	16 (55.2)	1
Incomplete evacuation	9 (52.9)	21 (72.4)	0.213
Nausea	10 (58.8)	6 (20.7)	0.012
Burning	8 (47.1)	14 (48.3)	1
Belching	9 (52.9)	15 (51.7)	1
Acid regurgitation	6 (35.3)	9 (31)	1
Epigastric pain	8 (47.1)	13 (44.8)	1

The initial number of samples was low for certain microbiota and biochemical tests due to lack of complete data.

**Table 2 nutrients-16-01228-t002:** Gastrointestinal (GI) disorders by time according to study group.

	Gluten Group (n = 17)			Placebo Group (n = 27)			∆Intergroup Analysis
GI Disorders	Basal	Final	*p*	Basal	Final	*p*	
Abdominal pain	2 (1–2)	1 (0–1)	**0.013**	1 (0.5–2)	0 (0–1)	**0.001**	**ns**
Bloating	1 (0–2)	1 (0–1)	**0.033**	2 (1–3)	1 (0–2)	**0.000**	**ns**
Flatulence	1 (1–2)	1 (0–1)	**0.008**	2 (2–3)	1 (0–2)	**0.001**	**ns**
Diarrhea	1 (0–1)	0 (0–0)	**0.008**	0 (0–2)	0 (0–0)	**0.004**	**ns**
Stomach sounds	1 (0–2)	0 (0–1)	**0.031**	2 (1–2)	0 (0–1)	**0.000**	**ns**
Reduced consistency	1 (1–2)	0 (0–1)	**0.003**	1 (0–2)	0 (0–0)	**0.001**	**ns**
Constipation	0 (0–1)	1 (0–1)	0.305	0 (0–1.5)	0 (0–1)	0.177	**ns**
Urge to defecate	1 (0–2)	0 (0–1)	**0.031**	1 (0–2.5)	0 (0–1)	0.056	**ns**
Incomplete evacuation	1 (0–2)	1 (0–2)	0.886	1 (0.5–2)	1 (0–1)	**0.033**	**ns**
Nausea	1 (0–2)	0 (0–0)	**0.021**	0 (0–0)	0 (0–0)	0.125	**ns**
Burning	0 (0–1)	0 (0–1)	0.480	0 (0–1)	0 (0–1)	0.109	**ns**
Belching	1 (0–2)	0 (0–1)	**0.021**	1 (0–1)	0 (0–1)	0.388	**ns**
Acid regurgitation	0 (0–1)	0 (0–0)	0.375	0 (0–0.5)	0 (0–0.5)	1	**ns**
Epigastric pain	0 (0–2)	0 (0–0)	0.068	0 (0–1)	0 (0–0)	**0.017**	**ns**
Depositions	3.24 ± 2.02	2.76 ± 1.79	0.290	3 (2–6)	3 (1–4.5)	0.454	**ns**
Bristol stool scale	4.24 ± 1.09	3.29 ± 1.53	**0.016**	4 (4–5)	4 (2.5–4)	**0.047**	**ns**

All bold show significant data. ns: not significant.

## Data Availability

Most of the data that support the findings of this study are available within the article and its Appendix A files or from the corresponding author upon reasonable request. Some of the data are not publicly available due to their containing information that could compromise patient privacy.

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
