# Peer review of "Effects of Gluten on Gut Microbiota in Patients with Gastrointestinal Disorders, Migraine, and Dermatitis"

_nutrients, 2024, doi:10.3390/nu16081228_

Round 1
Reviewer 1 Report
Comments and Suggestions for Authors
Figures
Figures 2, 3 and 4 are fairly low resolution- perhaps this is from compression into the pdf file, but please improve if possible.
Figure 5 – Please change the color of the delta group, as the yellow is hard to see.
Introduction
Line 41 – The sentence reads “The gluten-free diet’s symptoms are characterized…”, but it seems that this is meant to read “Symptoms of gluten intolerance” or “symptoms that the GFD aims to overcome include”. Please clarify.
The introduction should include a clear statement of the hypothesis. It would also be helpful if there was a paragraph reviewing what is already known about the gut-brain axis and gluten as an inflammatory modulator. Without clearer presentation of this information, it is not obvious why this study is necessary to advance current understanding of this condition.
Materials and Methods
Please include more information about inclusion and exclusion criteria for the study population. How was GID classified and validated? Did any of the participants have celiac disease? Had any of the participants been previously adhering to a gluten-free diet?
Was the study blinded or double blinded?
Line 85 – “100 g of wheat/day, GFD, or placebo” makes it sound like placebo and GFD are two different treatments, but I don’t think they are? Please clarify in the text.
Please provide a more robust description of how and when blood and stool samples were collected.
Results
Lines 215-216 – p=0.058 does not constitute statistical significance according to your standards listed elsewhere in the paper, so why is it being reported that there was a significant reduction in Chao1 diversity in the gluten group?
Lines 215-223 – Please be cautious of sentence structure and where you add the references to statistical analyses. It’s extremely hard to follow this paragraph when comparing to Figure 2. For example, “Placebo group increased the Chao 1 and gluten group decreased (n=24, p=0.011)”, this reads as if p=0.011 refers to the decrease in Chao1 for gluten, but we can see on the graph that the p value for the gluten group’s Chao1 is p=0.058.
Line 267 – How similar is “very similar”?
I think it’s really important to report on whether the people in the study had been following a gluten-free diet before the study. Without this information, there is no frame of reference for the differences observed in GI symptoms. For example, if an individual was already saying they have a gastrointestinal disorder but they had been continually eating gluten, they might not see differences in symptoms when taking the gluten sachet whereas someone who had been on a gluten-free diet previously might. This would be important to report on in the data and if there was variation in the participants gluten consumption beforehand, I wonder if that could have washed out some potentially interesting results from this study.
This study would be improved substantially by some kind of analysis that combines the gut microbiome data with the symptom survey data. I recommend adding that to improve the novelty of this paper. Without it, the findings are a bit disparate and I’m not sure that they advance our current understanding of these conditions in a meaningful way.
Comments on the Quality of English LanguageThere are some sentences that are unclear, but they could easily be fixed with a thorough proofreading.
Reviewer 2 Report
Comments and Suggestions for Authors
In the manuscript of Oliva et al, the authors aim to identify the impact of gluten on gut microbiota of patients with gastrointestinal disorders, migraine and dermatitis. The strategy the authors take to identify glutens impact is interesting, but there are several issues that need clarification.
1. The recruitment of patients has to be in details explained. Selection and excluding criteria as well dietary habits of the participants. Has been gluten part of the participants diet also has to be clarified. All this are essential as can impact the final results observed in the study.
2. It is very misleading the use of placebo and gluten-free diet. The reviewer understands that both are the same condition in the current study. To avoid confrontation, explain in the Material and Methods that placebo is gluten-free diet and further in the text use only placebo. Otherwise, the discussions and conclusions can be misleading.
3. The authors select 8 g gluten per day. What is the base for this selection? The daily amount of gluten is usually higher than 8g. The lack of significant differences in the microbiota could be also driven by the low daily intake of gluten, but also the short treatment. All this need to be scientifically justified in the Material and Methods part and discussed later in the Discussion part.
4. The Biomarker results are all respective supplementary results are not accessible for the reviewer, therefore cannot complete revise the manuscripts.
5. In the Material and Methods part, the library preparation not completely explained. Please include the primers for V4 16SrRNA and ITS1/ITS2 amplification and the respective conditions for PCR and purification.
6. Include, supplementary tables that include all microbiota data and the respective statistical analyses.
7. Increase the font size of all Figures and improve their quality.
